



# Effect of vegetation distribution driven by hydrological fluctuation on sedimental stoichiometry regulating N$_2$O emissions in freshwater wetland

Huazu Liu[1,4], Qiu Jin[2,5], Ruijie Shi[3], Chengxu Lv[3], Junxiao Luo[1,4], Yan He[1,4], Wei Yang[1,4], Xiaoguang Xu[3], Shenhua Qian[1,4], and Wei Li[1,4]

[1]Key Laboratory of the Three Gorges Reservoir Region's Eco-Environment, Ministry of Education, Chongqing University, Chongqing 400045, China
[2]State Key Laboratory of Hydrology-Water Resources and Hydraulic Engineering, Nanjing Hydraulic Research Institute, Nanjing 210029, China
[3]School of Environment, Nanjing Normal University, Nanjing 210023, China
[4]Department of Ecological Sciences and Engineering, Chongqing University, Chongqing 400045, China
[5]College of Agricultural Science and Engineering, Hohai University, Nanjing 210098, China

**Correspondence:** Wei Li (liweieco@cqu.edu.cn)

**Abstract.** Hydrological conditions drive the distribution of plant communities in wetlands to form vegetation zones where the material cycling varies with plant species. This mediation effect caused by the distribution of vegetation under hydrological conditions will affect the emission of N$_2$O during the nitrogen migration in wetlands. In this study, five vegetation zones in the second largest wetland of China were investigated *in situ* during high and low water levels to elucidate the effect mediated by

vegetation. With the increase in the rate of change of water levels, the zones of the mud flat, nymphoides, phalaris, carex, and reeds were distributed in sequence in the wetland, and the densities of carbon and nitrogen sequestrated by plants also increased. The carbon and nitrogen densities in each zone during low water level was significantly higher than that during high water level, while the organic carbon and the total nitrogen of sediments during high water level was higher. Sediments converted between source and sink for both carbon and nitrogen, during the annual fluctuation in water level. The flux in N$_2$O emissions showed

significant differences between the vegetation zones during each water level period. The emission flux decreased with the increasing C:N ratio in sediments, approximating the threshold at 0.23 $\mu$g m$^{-2}$ h$^{-1}$ when the C:N ratio > 25. The phylum abundance of Firmicutes, Proteobacteria, and Chloroflexi in sediments increased with flooding. The denitrifying *nirS* and *nirK* genes and anammox *hzsB* gene were significantly affected by water level fluctuation, with the maximal variations of these genes occurring in the mud flat and nymphoides zone. The results indicate that the distribution of plants under hydrological

conditions modified the stoichiometric ratio of sediments, resulting in the variations of N$_2$O emission fluxes and microbial communities in vegetation zones. Therefore, hydraulic regulation rather than direct planting would be an effective strategy to reduce greenhouse gas emissions in freshwater wetlands.





## 1 Introduction

Hydrology is the most influential factor regulating the structure and function of lakeshore zones, which can be drastically altered

by anthropogenic activities, such as the construction and operation of dams and reservoirs (Mitsch and Gosselink, 2007; Sun et al., 2018). In the hydrological rhythm caused by human activities, reservoir impoundment upstream is a critical driving factor, which significantly alters the natural hydrological regimes of downstream rivers and river-connected lakes (Barnett and Pierce, 2008; Gao et al., 2014; He et al., 2018). This can affect basic ecological functions such as the succession of plant and animal communities, carbon transfer, nitrogen cycle, and biodiversity (Bartle, 2002; Brown et al., 2014; Mcclain et al., 2014).

Moreover, vegetation as the key element of wetlands affects the habitats of other organisms (Brix, 1997). Thus, establishing a population of vegetation is a key factor in the lakeshore zone, and it must be studied for its response to hydrological rhythms.

   The hydrological conditions are generally considered to be the dominant factor that determines vegetation stratification (Tan et al., 2016; Toogood et al., 2008; Wilcox and Nichols, 2008). A significant ecological response relationship exists between the variation of hydrological conditions and the stratification of vegetation (Todd et al., 2010; Toner and Keddy, 1997). Many

studies have shown that wetland vegetation types are related to hydrological factors such as water level, flooding depth, and flooding time (Robertson et al., 2018; Hu et al., 2015; Hebb et al., 2013). Generally, the flooding depth modifies the moisture content of wetland soil, which leads to variations in the area and distribution of wetland vegetation. The discrepancy in the tolerance of vegetation to floods and droughts of soil and the time and frequency of flooding result in different distributions of plants (Asaeda, 2008; SimonCollison, 2002). Moreover, the different pathways of material transfer of sedimentary carbon and

nitrogen in vegetation zones are caused by the traits of species, including growth rate, bioavailability, and decomposition rate of plant residues (Deegan et al., 2011; Trimmer et al., 2012). The effect of hydrology is not only reflected in the stratification of vegetation but it also indicates that the short-term alternation of wetting and drying in wetland will affect the withering and fall and growth of plants (Wang et al., 2014; Iwanaga and Yamamoto, 2008). This further alters the content of the physio-chemical conditions in the sediment.

Wetland is the largest component of the terrestrial biological carbon pool (Chmura et al., 2003), and the sedimentary organic matter in wetland originates from vegetation and sediment transport. Plants rely on photosynthesis to capture $CO_2$ in the atmosphere (Ouyang and Lee, 2013), and organic matters enter the sediment with the fall and death of plants (Bouillon et al., 2000). In contrast, the sediments provide a huge number of elements for plant growth, such as nitrogen, phosphorus, and potassium (Kochian et al., 2004). Due to differences in the biomass and properties of vegetation, the sediments harvest a diverse

content of carbon and nitrogen. When the climate and soil parent material are the same, the content of carbon and nitrogen in the sediments is mainly affected by vegetation (Bull et al., 1999; De et al., 2008; Gao et al., 2012). Thus, studies are needed to determine how sediment, as the source and sink of these elements, is affected by the distribution and decline of vegetation, to reveal the ecological response between vegetation and sediment under hydrological stress. Furthermore, organic matter and the total nitrogen in soil are the main components of the terrestrial carbon and nitrogen pool, which reflects the level of soil quality,

with the relationship between them being expressed by the carbon to nitrogen (C:N) ratio. The organic matter in soil contains a large amount of carbon and nitrogen and the activities of microorganisms require carbon as an energy source. The C:N





ratio of soil significantly affects the rate of microbial decomposition of organic matter and the rate of nitrogen mineralization (Springob and Kirchmann, 2003). Thus, the C:N ratio indicates the balance of carbon and nitrogen and the decomposition degree of organic matters in sediment (Huang et al., 2007; Elser et al., 2003).

Nitrous oxide ($N_2O$) is a kind of typical greenhouse gas, with a potential greenhouse effect that is 298 times that of $CO_2$ (Church et al., 2013). Inland waters are considered to be significant components of the global $N_2O$ budget (Ravishankara et al., 2009; Jurado et al., 2017). Furthermore, the riparian zone, the transitional boundary between terrestrial and aquatic ecosystems, has been suggested to have an important role in removing nitrogen (N) pollutants through microbial N-cycle pathways (Kim et al., 2016). Thus, the riparian zones are also regarded as hotspots for $N_2O$ production in natural ecosystems (Groffman et al.,
2000; Wang et al., 2006). From the perspective of vegetation distribution caused by long-term hydrological conditions, the root systems and litters of plant species not only modify the physio-chemical conditions of the soil (Zhang et al., 2010), but they also provide discrepant organic matter for the activities of microorganisms (Zhai et al., 2013), thereby affecting the emission of $N_2O$. Due to the different anatomical structures and physiological characteristics of different vegetations, the transport capacity of vegetation to $N_2O$ also differs (Zhang et al., 2011). In addition, short-term dry and wet alternations strongly affect soil carbon
and nitrogen cycles and microbial activities, thereby affecting $N_2O$ emissions (Cui and Caldwell, 1997; Manzoni et al., 2014).

Four reduction steps including nitrate ($NO_3$-) reduction, nitrite ($NO_2$-) reduction, nitric oxide (NO) reduction, and $N_2O$ reduction together form denitrification, which is the main way to produce $N_2O$ in water (Kuypers et al., 2018; Bateman and Baggs, 2005). As a microbial process to produce $N_2O$, denitrification in riparian zones is regulated by many environmental factors, such as temperature, concentration of NO3-, moisture, pH, and carbon availability (Burgin et al., 2010; Groffman and
Crawford, 2003; Xiong et al., 2015). Moreover, organic carbon is used as the electron donor for heterotrophic denitrification processes (Li et al., 2019). For agricultural soils, wheat straw addition enhances $N_2O$ emission, with greater emissions being observed from the narrow C:N ratio amendments (Frimpong and Baggs, 2010; Li et al., 2013). These further imply the significance of studying the effect of the C:N ratio of sediments on the flux of $N_2O$ emissions. Moreover, since most lake and reservoir sediments are in an anaerobic environment, denitrification is the most important way to produce $N_2O$ in these water bodies.
As another key microbial process for nitrogen removal, the anammox which oxidizes $NH_4$+ and reduces $NO_2$- to produce $N_2$ under anaerobic conditions has attracted wide attention in recent years (Kartal et al., 2007; Strous et al., 1998). In wetlands, both anaerobic denitrifying bacteria and anammox bacteria may be affected by water level, and both use NO2- as the reaction substrate. Nevertheless, the different denitrification and anammox functioning genes in different vegetation zones are rarely reported to be caused by water conservancy conditions, and more researches are needed to understand these two processes in
wetland.

In this study, five vegetation zones (reed, carex, phalaris, nymphoides, and mud flat) in the lakeshore zone of Dongting Lake were investigated during high and low water levels. The objectives of the study were to examine (1) the relationship between changes in the hydrological conditions and the stratification and decline of vegetation; (2) the ecological factors of sediments changed by vegetation and the effect of the C:N ratio of vegetation on the C:N ratio of sediments; (3) the differences in $N_2O$
emission fluxes and the C:N ratio of sediments as the dominant factor for the $N_2O$ emission flux; and (4) the differences in microbial communities and functional genes for nitrogen cycle.





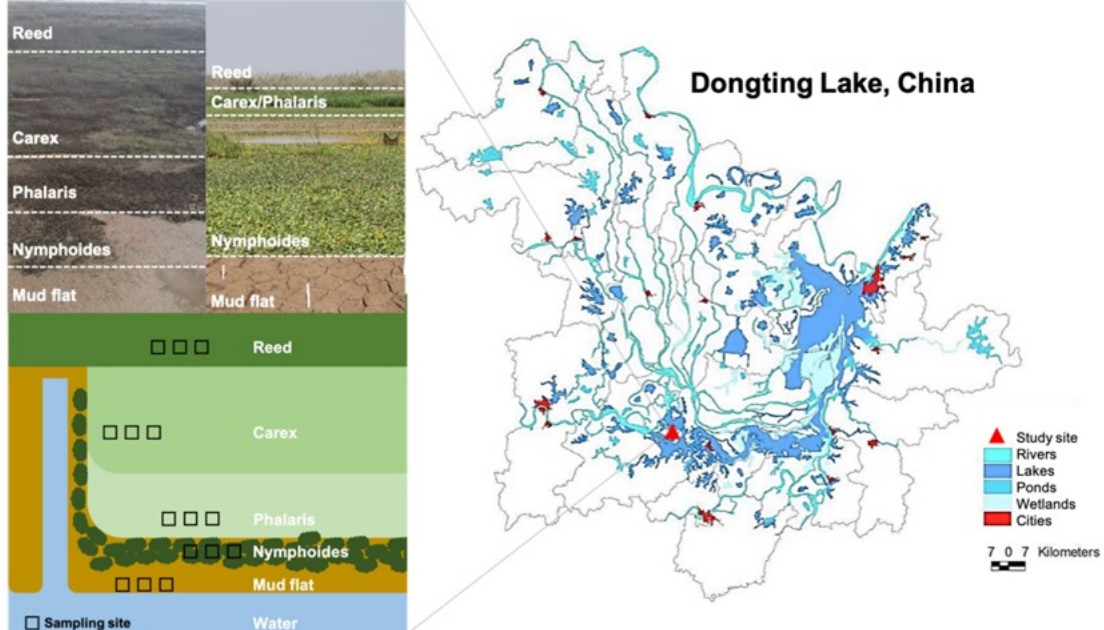

**Figure 1.** Location and distribution of sampling sites in Dongwa wetland of Dongting Lake, China (three sampling sites were used in each vegetation zone)

## 2 Materials and methods

### 2.1 Study area and data sampling

This study was conducted in Dongting Lake, which is in Hunan Province, China. The lake is the second-largest freshwater lake in China, with an area of about 2625 $km^2$. It still maintains a natural connection to the Yangtze River, which is of great significance for maintaining the integrity of the complex ecosystem of the middle and lower reaches of the Yangtze River. Located in the subtropical monsoon climate zone, the lake region's wet season is between July and September, while its dry season is between November and the following February (Yuan et al., 2015). The hydraulic conditions of Dongting Lake appear to undergo periodic changes under the influence of the Three Gorges Reservoir (TGR). As an important habitat for protecting waterfowl and waders, the Dongting Lake wetland has been registered in the Ramsar Convention.

The study area (28°52'24" N, 112°13'57" E) is located in the Dongwa wetland of West Dongting Lake. The study area was divided into five zones (four vegetation zones and a mud flat; Fig. 1). The dominant species in each vegetation zone are Phragmites australis (reed), Carex cinerascens (carex), Phalaris arundinacea (phalaris), and Nymphoides peltatum (nymphoides), and no vegetation occurs in the mud flat. The elevations of the vegetation zones are as follows: reed > carex > phalaris > nymphoides > mud flat.



All samples were collected in October 2019 (low water level) and June 2020 (high water level), and in triplicate for each vegetation zone. The depth ratio was recorded by a self-calculating water level depth monitor (Onset HOBO U20-001-02) from October 2019 to October 2020 and the quadrats of plants (1 m × 1m) were set in each vegetation zone. All of the above-ground plants in the sample were collected to measure the above-ground biomass of each vegetation. The surface layer (0-10 cm) sediments were collected in each vegetation zone.

## 2.2 Elemental analyses

A subsample of dry plant material was washed and dried again, and then ground to a fine powder using a ball mill before chemical analysis. Total organic carbon (C) and total nitrogen (N) concentrations were determined by thermal combustion using an elemental analyzer (CE Instruments model NA2500 Nitrogen Carbon Analyzer) and expressed as percent of dry weight. Finally, the C:N ratio, carbon density, and nitrogen density of the vegetation were calculated.

## 2.3 In situ measurement of $N_2O$ emission flux

The closed-chamber technique was used to measure the in situ $N_2O$ emission rate according to Wang (Wang et al., 2006). Three replicated chambers were used at each site during high and low water levels. A static chamber with an upper chamber and a pedestal was used to measure the soil-atmosphere $N_2O$ flux during low water level. The upper chamber was equipped with an air fan to mix the air and a temperature probe to detect the temperature during the determination. The pedestal was inserted into the soil for about 10 cm with the gutter filled with water to form a water seal. For water-atmosphere $N_2O$ flux during high water level, a pedestal with a water wing was used to keep the chamber floating on the water surface. During the determination, 150 mL gas samples were taken at 0-10 min intervals from the headspace of the chamber using polytetrafluoroethylene tubes and glass syringes and then injected into 200 mL multi-layer foil sampling bags. The gas samples were immediately transported to the laboratory and measured within one day by gas chromatography (Agilent 7890A) equipped with an electron capture detector. The $N_2O$ emission rate was calculated from the linear change in concentration with time.

## 2.4 Quantitative PCR and high-throughput sequencing of microbial community

During high and low water levels, about 5 g of the surface layer (0-10 cm) sediments were retained from each vegetation zone to determine the bacterial community structure and abundance. The microbial community structure in sediments was detected using Illumina MiSeq sequencing of the 16S rRNA gene with the primers *515FmodF* and *806RmodR* (Walters et al., 2016; Sampson et al., 2016). Denitrifying bacteria were quantified by real-time quantitative PCR (ABI 7500). The denitrifying functional genes (*nirS* and *nirK*), were amplified using primers *cd3aF* (5'-GTSAACGTSAAGGARACSGG-3') and *R3cdR* (5'-GASTTCGGRTGSGTCTTGA-3') (Palmer et al., 2011), *FlaCu* (5'-ATCATGGTSCTGCCGCG-3') and *R3Cu* (5'-TTGGTGTTRGACTAGCTCCG-3') (Throbck et al., 2004), respectively. The anammox functional genes (*hzsB*), were amplified using primers *AMX818F* (5'-ATGGGCACTMRGTAGAGGGGTTT-3') and *AMX1066R* (5'- AACGTCTCACGACAC-GAGCTG -3') (Tsushima et al., 2007).





## 2.5 Statistical analysis

One-way analysis of variance (ANOVA) was used to test the statistical significance of differences of ecological factors in sediments between high and low water levels. A *t*-test analysis was used to compare the differences between the $N_2O$ emission

flux, and between the functional gene abundances. The linear regressions were used to assess the effects of hydrological conditions on biomass of vegetations, traits of vegetations on the ecological factor in sediments, and the ecological factors in sediments on the $N_2O$ emission flux. The exponential regression was used to assess the effects of the C:N ratio in sediments on the $N_2O$ emission flux. All statistical analyses were carried out using SPSS (v22.0, SPSS Inc., North Chicago, IL, USA). The level of significance was $P < 0.05$ for all tests.

# 3  Results

## 3.1  Variation of water level in vegetation zones

The low water level period of Dongting Lake lasted from October 2019 to May 2020 during the one-year water level monitoring, and the high water level period of Dongting Lake was from May 2020 to November 2020 (Fig. 2a). The water level in each vegetation zone reached its maximum on June 25, 2020 (mud flat: 3.712 m, nymphoides zone: 3.710 m, phalaris zone: 3.687

m, carex zone: 3.559 m, and reed zone: 2.632 m). The change rate for the water level in each vegetation zone was calculated according to the ratio of the maximum water level to the elapsed time to reach the maximum. The rapid change in water level generally indicates the relative great effect of hydrological changes (time and depth of flooding) on plants, and vice versa. The change rate of the water level in each zone had a significantly negative correlation with carbon and nitrogen density of the vegetations ($P < 0.05$) (Figs. 2b and 2c). Reeds with developed root and stout stem grow well in the zone with slow change

rate of water level, assimilating a large amount of carbon ($326.738 \pm 9.623$ g m$^{-2}$) and nitrogen ($16.413 \pm 0.403$ g m$^{-2}$). Nymphoides, phalaris, and carex zones were more affected by hydrological changes, where the plant biomass was less than that of the reed zone. The correlation between water level variation and plant assimilation indicated that the long-term change of hydrological regime induced the stratification of vegetation.

## 3.2  Characteristics of sediment in vegetation zones

The maximal TOC concentration was observed in sediments of the phalaris zone during low water level ($42.743 \pm 0.384$ g kg$^{-1}$), and the highest concentrations were observed in sediments of the nymphoides zone during high water level ($66.710 \pm 1.924$ g kg$^{-1}$) (Table S1). Similarly, the highest concentrations of TN were observed in sediments of the phalaris zone during low water level ($1.617 \pm 0.023$ g kg$^{-1}$), and in sediments of the nymphoides zone during high water level ($3.304 \pm 0.062$ g kg$^{-1}$). The concentrations of TOC in the sediment of carex and nymphoides zones and mud flat at high water level were significantly

higher than that during low water level ($P < 0.05$). The concentrations of TN, $NH_4^+$-N, and $NO_3^-$-N in the sediments of each vegetation zone during high water level were significantly higher than that of the low water level ($P < 0.05$). During the low water level, the C:N ratio of sediments decreased as the elevation decreased and the C:N ratio of sediments in the reed zone





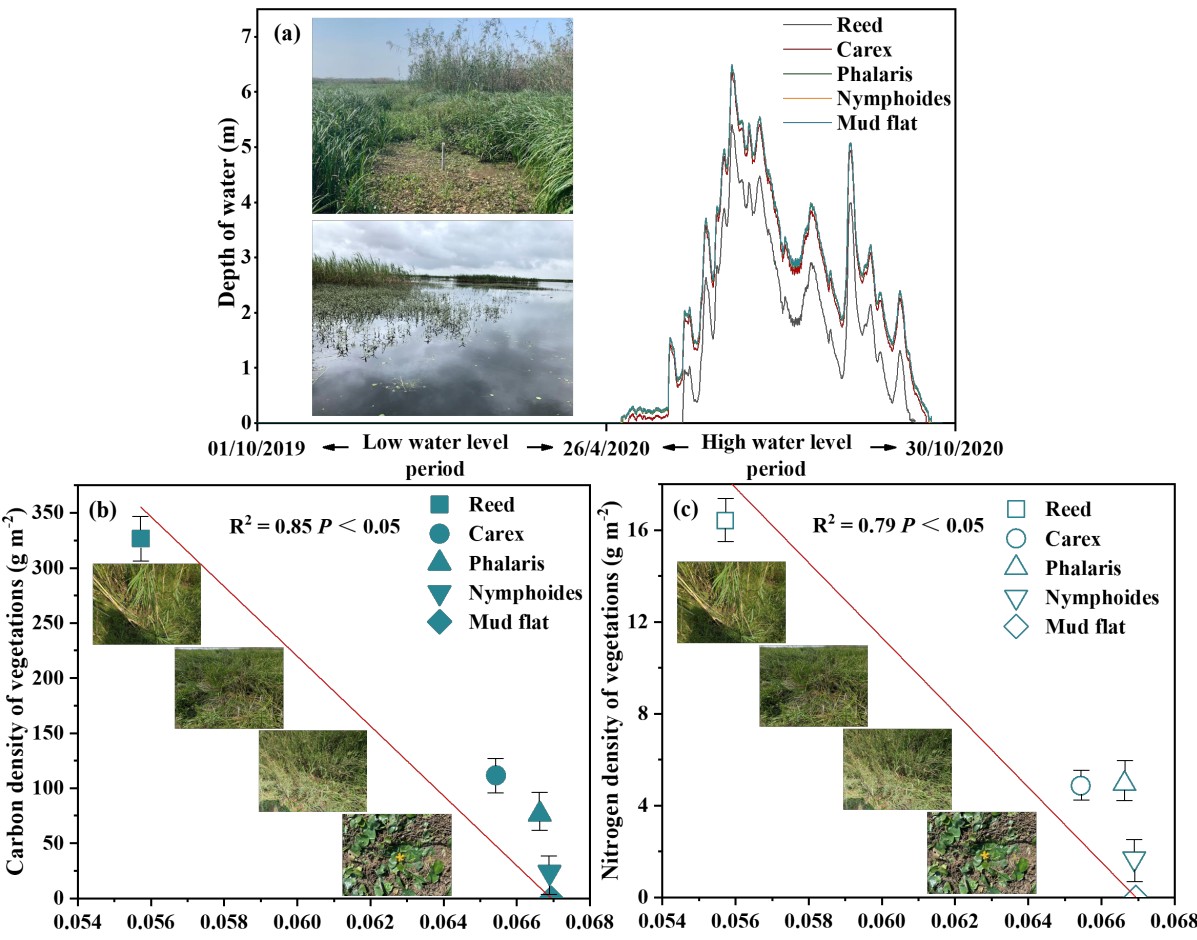

**Figure 2.** Carbon density and nitrogen density of vegetation under hydrological stress in sampling sites. (a) Change in water level during the one-year monitoring in the different vegetation zones. (b) Relationships between carbon density of vegetation and change rate of water level. (c) Relationships between nitrogen density of vegetation and change rate of water level. )





was the highest (50.802 ± 0.847). During high water level, the highest C:N ratio of sediments was observed in carex zone sediments (26.542 ± 2.274). Except for the mud flat zone, the C:N ratio of sediments in the other four vegetation zones during low water level were significantly higher than those during high water level ($P < 0.05$). The content of varying factors differed during different water level periods, implying the influence of the alternation of dry and wet lakeshore zones on the sedimentary physical and chemical environment during a period of water level change.

### 3.3 Feedback between sediment and plants in the vegetation zones

The carbon density of each vegetation during low water level was significantly higher than that during high water level ($P < 0.05$) (Fig. 3a). In contrast, the concentration of TOC in the sediments of each vegetation zone during high water level was higher (Fig. 3b). For the nitrogen density of vegetation, each vegetation during low water level was significantly higher than that during high water level ($P < 0.05$) (Fig. 3c). In contrast, the concentration of TN in the sediments of each vegetation zone during high water level was higher than that during low water level (Fig. 3d). The C:N ratio of sediments and the C:N ratio of vegetation showed a significant positive correlation during low water level ($P < 0.05$), and showed no significant correlation during high water level ($P > 0.05$) (Fig. S1). This showed the role of sediment as sources and sinks during different water levels. During high water level, the withering and fall of vegetation led to the input of organic matter into the sediment, which became a sink of organic matter. During low water level, the growth of vegetation absorbed a lot of elements from the sediment, which became a source of these elements. The content of carbon and nitrogen in the sediment was affected by vegetation.

### 3.4 N₂O emission flux in vegetation zones

N₂O emission flux ranged from 0.175 ± 0.054 to 0.561 ± 0.067 $\mu$g m$^{-2}$ h$^{-1}$ during high water level, and ranged from 0.197 ± 0.019 to 0.988 ± 0.267 $\mu$g m$^{-2}$ h$^{-1}$ during low water level (n = 3) (Fig. S2 and 4a). The N2O emission fluxes in the nymphoides zone and mud flat during low water were significantly higher than that during high water level ($P < 0.05$). In contrast, the N₂O emission flux in the reed zone during low water was significantly lower than that during high water level ($P < 0.05$). The N₂O emissions flux showed significant differences between vegetation zones during each water level period ($P < 0.05$) (Fig. 4a). This showed that the N₂O emission fluxes in the vegetation zones were varied, and differences were seen in the varied water level periods. Moreover, a significantly negative correlation occurred between N₂O emission flux and the concentrations of TOC in sediments during low water level ($P < 0.05$). Nevertheless, no significant correlation was seen during high water level ($P > 0.05$) (Fig. S3a). In each water level period, no significant correlation was found between the N₂O emission flux and the concentrations of TN in sediments ($P > 0.05$) (Fig. S3b). A significant negative correlation was observed between the N₂O emission flux and the C:N ratio of sediments ($P < 0.05$) (Fig. 4b). Moreover, when the C:N > 25, the N₂O emission flux gradually stabilized (0.23 $\mu$g m$^{-2}$ h$^{-1}$). This suggested that the C:N ratio of sediments was the dominant factor for the N₂O emission flux, and C:N = 25 may be the threshold value for the increase in the N₂O emission flux.



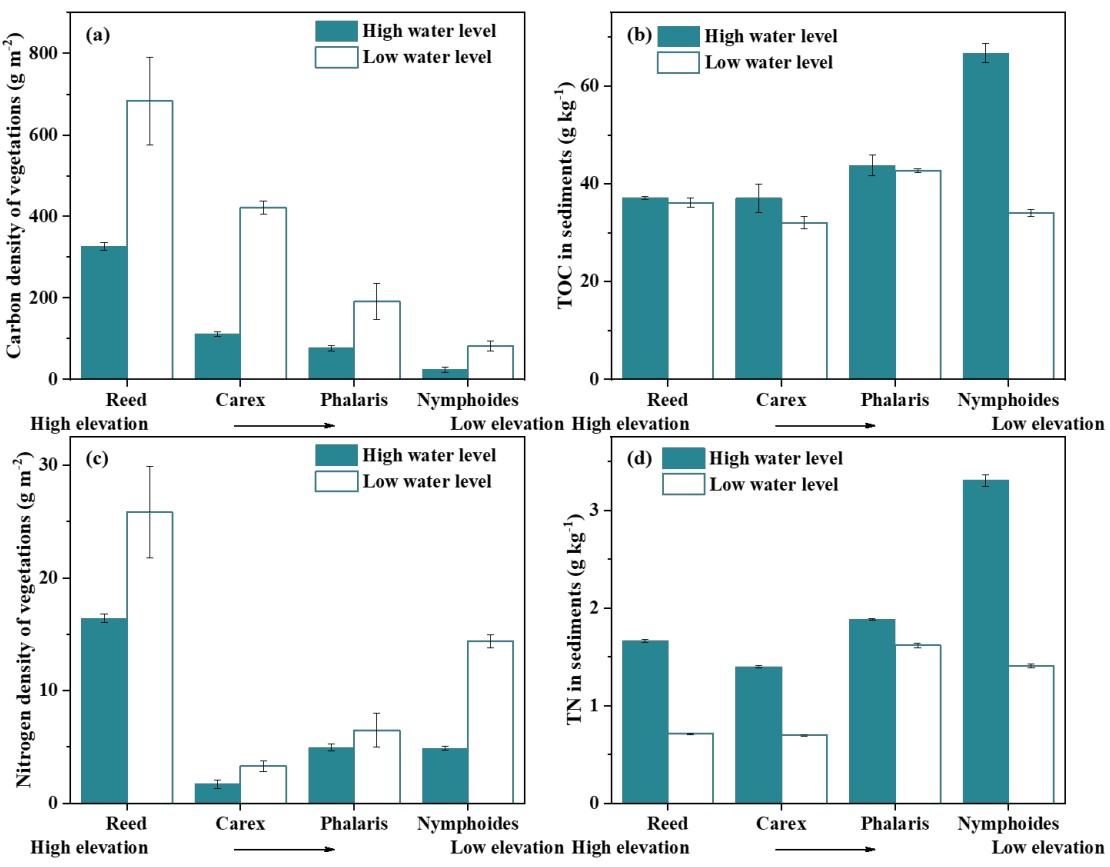

**Figure 3.** Content of carbon and nitrogen in vegetation and sediments during different water levels. Carbon (a) and nitrogen density (b) of vegetation in different vegetation zones. Concentration of TOC (c) and TN (d) in sediments in different vegetation zones

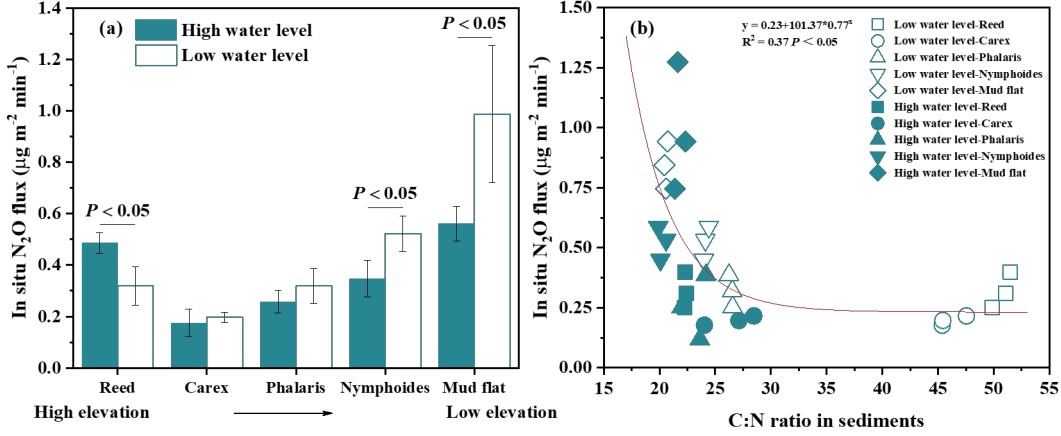

**Figure 4.** In situ N2O emission flux during different water levels (a), and the C:N ratio (b) in sediments during different water levels





## 3.5 Microbial communities and functioning bacteria in the vegetation zones

A total of 13009 OTUs were detected from the sediments at the end of the experiment. The OTUs in the sediments of the reed
zone (5539) and carex zone (6489) during high water level were richer in diversity than those in the sediments of the reed
zone (4864) and carex zone (6289) during low water level. In contrast, OTUs in the sediments of the phalaris zone (6419),
carex zone (6263), and mud flat (6330) during low water level were richer in diversity than those in the sediments of the
phalaris zone (6313), carex zone (5017), and mud flat (5279) during high water level (Fig. S4). During high water level, the
sediments of the vegetation zones shared the dominant phyla Proteobacteria (21.69-36.17%), Chloroflexi (6.82-13.53%), and
Actinobacteriota (6.75-19.66%). During low water level, the sediments of the vegetation zones shared the dominant phyla
Proteobacteria (10.09-27.71%), Chloroflexi (13.07-22.88%), and Actinobacteriota (6.52-30.83%) (Fig. 5a). The community
abundance of phyla Proteobacteria in the sediments of the vegetation zones, except for the carex zone during high water level,
were higher than that during low water level. The community abundance of phyla Chloroflexi in sediments of vegetation zones
during high water level were lower than that during low water level. For phyla Actinobacteriota, the community abundance in
sediments of vegetation zones, except for the reed zone, increased during high water level (Fig. 5b). Furthermore, the Shannon
diversity index of bacterial OTUs increased in sediments of the reed zone (6.76) and carex zone (7.17) during high water level,
compared to that in the sediments of the reed zone (5.78) and carex zone (7.12) during low water level (Fig. S4). There were
50 common species of microorganisms at the 'phylum' level in the sediments of all vegetation zones in the wet season and 55
in the dry season. There were 50 common phyla of microorganisms in the sediments of vegetation zones during high water
level and 55 common phyla during low water level (Fig. 5c). In each vegetation zone, the common phyla during the two stages
were: phalaris (61) > carex (59) > mud flat (56) > nymphoides (55) > reed (53) (Fig. 5d). The dominant community abundance
in terms of phylum and the Shannon diversity index of bacterial OTUs varied during high and low water levels. This showed
that the microbial community was affected by the water level.

During high water level, the sum of *nirS* and *nirK* genes ranged from $(4.27 \pm 1.53) \times 10^8$ to $(6.76 \pm 0.11) \times 10^8$ copies
g-1 dry sediment (nymphoides > reed > mud flat > phalaris > carex). During low water level, the sum of *nirS* and *nirK* genes
ranged from $(8.10 \pm 0.46) \times 10^7$ to $(5.76 \pm 0.82) \times 10^8$ copies $g^{-1}$ dry sediment (carex > reed > phalaris > nymphoides > mud
flat). The sum of *nirS* and *nirK* in sediments of each vegetation zone showed a significant difference between the two water
level periods ($P < 0.05$) (Fig. 6a). The sum of these genes showed significant differences between vegetation zones during each
water level period ($P < 0.05$). Correspondingly, the hzsB gene ranged from $(9.95 \pm 0.03) \times 10^7$ to $(3.06 \pm 0.09) \times 10^8$ copies
$g^{-1}$ dry sediment (nymphoides > phalaris > reed > mud flat > carex) at the high water level and ranged from $(1.07 \pm 0.03) \times 10^7$ to $(1.38 \pm 0.03) \times 10^8$ copies $g^{-1}$ dry sediment (reed > carex > phalaris > nymphoides > mud flat) at the low water level
(Fig. 6b). The content of $N_2O$-producers and anammox varied with the vegetation zones, which was affected by changes in
water levels.





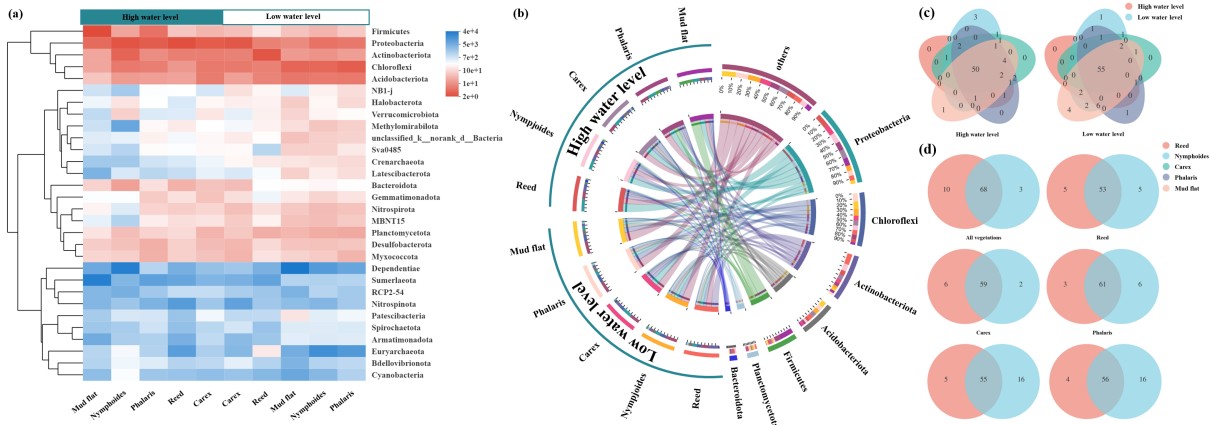

**Figure 5.** Effect of water level on bacterial communities (a) and dominant species (b) in different vegetation zones. Similarities of bacterial communities (c) and species (d) at the phylum level

## 4 Discussion

### 4.1 Long-term effect of hydrological conditions on vegetation stratification

The hydrological rhythm and biogeochemistry in the upstream area of this lake had undergone marked changes since the impoundment of the TGR (Zhang et al., 2012; Wu et al., 2016). Dongting Lake, located downstream of the TGR, still maintains a natural connection with the Yangtze River, where hydrological fluctuations of the lake have been affected by the operation of the TGR. The TGR watershed generally fluctuated between a base level of 145 m during the wet season (May to September) and a peak level of 175 m during the dry season (October to April) (Bao et al., 2015). The annual hydrological changes in Dongting Lake were contrary to those in the TGR (Fig. 2a). Moreover, the impact of the TGR operation on the hydrological rhythm of Dongting Lake was confirmed by building a river-lake model, where the ecological processes such as nitrogen cycling in the lake were profoundly affected by anthropogenic regulation in hydrology (Li et al., 2020).

Hydrological conditions were the key factors affecting species diversity, and the stability and distribution of the wetland plant community, especially at a local scale (Auble et al., 2005; Castelli et al., 2000; Boar, 2006). Long-term fluctuations in water level affected the moisture of the soil in the wetland, which directly led to changes in the habitat for specific species. The varied tolerance of plants to the alternation between dry and wet states, which were driven by hydrological conditions, led to the succession and distribution of vegetation. As a result, the carbon and nitrogen densities of vegetation were affected by the water level change rate that reflected hydrological conditions such as flooding time and depth (Figs. 2b and 2c). The reed zone had the shortest flooding time, while the biomass of the reeds was the largest among all types of vegetation zones. No plants grew in the mud flat, due to the maximum flooding time and depth of the different zones. The duration of flooding events also drove the segregation of the plant communities, and the lowest biomass and species with the least richness developed in pots that were continuously flooded (Casanova and Brock, 2000). Moreover, the niche of plants reflected the tolerance to





disturbance and the ability to resist competition from species, to reveal the distribution of plants in habitats (Tilman, 1999).

The discrepancies between the niches of plant species caused by hydrological conditions indicated the essence of stratification of vegetation zones. Specifically, the diversity of plant species was negatively correlated with the variability of hydrology in Poyang Lake, which is another large lake connected to the Yangtze River (Xu et al., 2015). Because of the physiological structure (e.g., organs, tissues, and metabolism), nymphoides, as a floating-leaf root plant decompose the most quickly with the longest flooded time and the deepest flooded depth compared to the other four species. In contrast, emergent aquatic plants

such as reeds were least affected by the weak flooding.

## 4.2  Short-term effect of hydrological conditions on nitrogen cycling

Annual flooding had a dramatic impact on the physiological processes of plants such as the absorption and transportation of mineral nutrients, and the accumulation of organic matter, which restricts the plant growth (Wang et al., 2014). It also caused drastic changes in the habitat for plant growth, such as dissolved oxygen, redox potential, and element pools (Liu et al., 2014;

Nilsson and Svedmark, 2002). Compared to high water levels, plants sequestrated more carbon during low water levels (Fig. 3a). In contrast, sediment harvested more carbon during high water levels (Fig. 3b). Thus, the carbon was mainly accumulated in sediments under flooding conditions, whereas the plant growth used more carbon when the wetland emerged from water. The feedback of the nitrogen pool between plants and sediments to hydrological fluctuations was similar to that of carbon (Figs. 3c and 3d). These explained the role of sediments as the sink of both carbon and nitrogen during high water level.

Furthermore, compared to the aerobic state of the sediment without overlying water, anaerobic conditions in water-saturated layers significantly retard the decomposition of organic matters, allowing it to accumulate as peat (Clymo et al., 1998). The sediment in the vegetation zone became a sink of carbon and nitrogen from the plants' detritus, when long-term flooding caused the decay of plants. In turn, when the water faded, the sediment provided nitrogen and amounts of inorganic carbon for plant growth. Other than nitrogen in the soil as the main source for plant growth, the carbon assimilated by plants mainly originated

from $CO_2$ in the atmosphere, which resulted in a greater variation of nitrogen content than that of carbon in the sediments (Figs. 3b and 3d). Thus, during annual fluctuations in water level, sediment converts between source and sink for both carbon and nitrogen. Although the function for source and sink of sediment in freshwater ecosystems has rarely been reported, the alternation between source and sink of soil in high-productivity salt marsh has been seen (Zhang et al., 2021). Both carbon and nitrogen accumulated in the soil during flooding, while the plant growth assimilated carbon and nitrogen from soil during dry

periods in the marsh. As the producer in the ecosystem of wetlands, plants provided energy for consumers (i.e., wading birds, zoobenthos, aquatic animals, and other wildlife). The functional conversion of sediment was due to the status of plants under hydrological stress, to alter the flow of energy in the aquatic food chain.

Short-term flooding caused the wither and fall of plants and the input of organic matter into the sediment (Bouillon et al., 2000), regulating the balance of carbon and nitrogen in the sediment. Since the biomass, tissues, and mineral nutrients of plants

varied with species, the input and output of both carbon and nitrogen to the sediment was discrepant (Gao et al., 2012). Both the characteristics of species and their habitats modified the decomposition process of plants (i.e., hydrolysis of plant tissues, dissolution of mineral components and organic matter, and the diffusion of micro-particles) in wetlands (Yarwood, 2018).





Therefore, the types, decline, and growth of plants led to the discrepancy in the content of carbon and nitrogen in the sediment. The C:N ratio of sediments was not only a key factor for evaluating the quality of the sediments, but it also served as a practical

indicator of the balance of carbon to nitrogen in soils (Huang et al., 2007). It can also reflect the processes of accumulation and decomposition for organic matter and the kinetics of equilibria between them (Elser et al., 2003). When the C:N ratio of sediment was >25.0, organic matter was accumulated faster than it was decomposed (Wei et al., 2009). In contrast, when the C:N ratio in sediment was 12.0-16.0, organic matter was decomposed faster by microorganisms (Bui and Henderson, 2013). The C:N ratios in the sediments of the carex zone during high water level and the reed, carex, and phalaris zones during low

water level were > 25.0 (Table S1), indicating that the organic matter of sediments in these zones accumulated faster than it decomposed. The C:N ratio in sediment increased with an increasing C:N ratio in the four species during low water level, while the C:N ratio of the sediment was not affected by the plants during the flooding (Fig. S1). This implied that the effect of the C:N ratio of plants on sediments was varied by anthropogenic regulation for water level.

### 4.3    N$_2$O emission in hydrological-stressed wetlands

The traits of aquatic plants (e.g., physiological structure, morphology, and life history) varied the N2O emission in the wetland ecosystem. Regardless of the fluctuating water level, the N$_2$O emission fluxes varied with the species of plants (Fig. 4a). When aquatic plants grew, the oxygen released by photosynthesis ameliorated the conditions of the dissolved oxygen, redox potential, and pH, which affected the diffusion of N$_2$O from sediment to the overlying water. The root system altered the rhizospheric environment and the attached microbes by releasing small molecule exudates, which also affected the emission of N$_2$O (Zhai et

al., 2013; Zhang et al., 2010). Despite the periodic water levels, the N$_2$O emission fluxes of the nymphoides and reed zones were higher than those of the other two vegetation zones (Fig. 4a). In the nymphoides zone, the highest concentration of TOC in the sediment among the four zones facilitated the microbial denitrification to increase N$_2$O emission. Although the concentration of TOC in the sediment was not maximal in the reed zone, the well-developed aeration tissues of reeds (i.e., vascular bundle and stoma of leaves) can provide the channels for N$_2$O diffusion, intensifying the emission from sediments (Baruah et al.,

2012). For example, the N$_2$O emission flux in the area with reed growth was 14-times greater than the area without reeds in a riparian wetland, which highlighted the large contribution of reeds to N$_2$O emissions (Gu et al., 2015). In such a complex wetland with vegetation stratifications under hydrological stress, the C:N ratio in the sediment effectively predicted the N$_2$O emissions (Fig. 4b). The release of N$_2$O from water to the atmosphere was a dynamic process, and related to the hydrological conditions that varied with the vegetation zones. Both the dynamics of N$_2$O emission and the heterogeneity of vegetation in

lakes and reservoirs resulted in N$_2$O emissions being affected by multiple factors (Miya and Firestone, 2000; Stadmark and Leonardson, 2007; Wu et al., 2009). The content of carbon and nitrogen in sediments was modified by the plant species that participated in the heterogeneity of vegetation. Thus, the concentration of TOC and TN in sediment alone cannot reflect the N$_2$O emission (Figs. S3a and S3b). In denitrification, the N$_2$O generated changes in the form of the nitrogen species, where the organic carbon was the electron donor (Li et al., 2019). This implies that carbon and nitrogen in conjunction determine the

microbial denitrification. Moreover, the C:N ratio characterizing the stoichiometric balance of carbon and nitrogen, also links the origins of carbon and nitrogen as the substrates for denitrification. When the C:N ratio of the sediment was < 25.0, the





decomposition of organic matter in the sediment is greater than the accumulation (Wei et al., 2009). The potential for releasing available nitrogen increases with the decreasing C:N ratio during the decomposition of organic matter (Cleveland and Liptzin, 2007). Thus, the N$_2$O emission flux decreased with the increase of the C:N ratio in sediment (Fig. 4b). When the accumulation

of organic matter was greater than decomposition, the release of nitrogen tended to be stable. Meanwhile, the N$_2$O emission approached the threshold (0.23 $\mu$g m$^{-2}$ h$^{-1}$), while the C:N ratio did not regulate the emission (Fig. 4b). Studies on the addition of plants also verified the effect of the C:N ratio on N$_2$O emission. For agricultural soils, N$_2$O emissions were inversely related to the C:N ratios of crop residue, with wide ratio residues generating lower emissions (Li et al., 2013). The addition of wheat straw enhanced the N$_2$O emissions, with greater emissions being observed from the narrow change to the C:N ratio (Frimpong

and Baggs, 2010; Zhou et al., 2020). Similarly, the withered and fallen plants varied the C:N ratio in sediment, which led to the different C:N ratios in the different zones (Fig. S1). Therefore, the C:N ratio of soil rather than the content of carbon and nitrogen was a key factor for predicting the function for N$_2$O emission. An exploration of the threshold indicated that neither the addition nor decline of plants can infinitely inhibit the N$_2$O emissions.

## 4.4 Microbial denitrification in hydrological-stressed wetlands

Denitrification in sediment was affected considerably by hydrological changes (Song et al., 2010). Frequent and successive drying and flooding resulted in observable changes in the sedimentary physio-chemical conditions (e.g., dissolved oxygen, content of organic matter), and the regulating of microbial denitrification (Cheng et al., 2007). Moreover, the input and edibility of organic matter from plants to sediments varied with the plant species, which altered the availability of the carbon source to the heterotrophic denitrifiers (Tall et al., 2011; Audet et al., 2014; Xiong et al., 2015). The phylum abundance of Firmicutes

and Proteobacteria, which performed denitrification (Zhu et al., 2021), increased with flooding (Fig. 5a). The anaerobic conditions of the soil caused by flooding were also more suitable for the growth of denitrifiers. This implied that more microbes were participating in denitrification during high water level. Flooding not only regulated the denitrifiers but also modulated other microflora. The phylum abundance of Chloroflexi, Acidobacteriota, and Bacteroidota also varied with water level (Fig. 5b). As photosynthetic autotrophic bacteria, Chloroflexi was involved in the transformation of inorganic carbon in sediments,

which occurred in anaerobic areas of water where light can irradiate (Klatt et al., 2013). Thus, the decrease of transparency caused by flooding resulted in the reduction of the phylum abundance. The aerobic Acidobacteriota mainly grew around the rhizosphere and was affected by plants (Lee et al., 2008). Its abundance decreased in the anaerobic sediments resulted by the dormancy or decay of plants during high water level period. Bacteroidota, which can adapt to a variety of niches, degraded refractory organics and participated in the carbon cycle in litter decomposition (Thomas et al., 2011). This phylum abundance

in sediments thus varied with the vegetation zones and the water levels. In addition, the nymphoides as floating-leaf plants, contain low contents of cellulose and lignin, which requires less bacterial species to degrade these complex organics and various intermediate products during the degradation (Jeffries, 1990). Thus, the phyla and diversity of microorganisms in the carex and phalaris zones were higher than that of nymphoides (Figs. 5d and S4). The litter of reed was more difficult to be utilized by microorganisms (Castle et al., 2019), which resulted in the lower phyla and diversity of reed zone than that of carex and

phalaris (Figs. 5d and S4).





Denitrification was the only known pathway responsible for the production and reduction of $N_2O$ (Kelso et al., 1999). Denitrification contains four reduction steps to $NO_3^-$, $NO_2^-$, nitric oxide (NO), and $N_2O$ (Kuypers et al., 2018). The reduction of $NO_2^-$, as the first step responsible for gaseous product, is catalyzed by nitrite reductases encoded by *nirS* or *nirK* genes. The content of the *nirS* gene increased with water level, while the contents of the *nirK* gene in reed and carex zones decreased with

flooding (Fig. 6a). The inconsistency between *nirS* and *nirK* genes highlighted that the sum of *nirS* and *nirK* genes can explore denitrification in wetlands with strong fluctuations in water level. The differences in the sum of genes between vegetation zones showed the discrepancy in the denitrification caused by stratification of vegetation under hydrological stress. During flooding, the sum of *nirS* and *nirK* in the reed and nymphoides zones and the mud flat were relatively high, which conforms to the $N_2O$ emission fluxes in these zones (Figs. 4a and 6a). The concentration of TOC in the nymphoides zone was higher under

flooding conditions, which is suitable for the denitrifiers to use organic matter (Audet et al., 2014). Thus, the sum of the nirS and nirK genes in the nymphoides zone was abundant. Although the sedimentary TOC in the mud flat was lower than that of the nymphoides zone, the water-solid interface formed more anaerobic conditions without the oxygen-enriched macrophytes, thus facilitating the denitrification communities (Cheng et al., 2007). In addition, the sum of the *nirS* and *nirK* genes decreased with decreasing water levels (Fig. 6a). When the wetland emerged from the water, the moisture in the sediment decreased, while

the pore water in the sediments ran off gradually. This facilitated the permeation of oxygen from the water-solid interface into the sediment, which generated the sedimentary conditions that tended to be aerobic. In turn, this restricted the growth of the anaerobic denitrifying bacteria (Xiong et al., 2015). Anammox carrying *hzsB* gene was anaerobic autotrophic bacteria, which obtain chemical energy by oxidizing compounds (Schouten et al., 2004). Therefore, the concentration was higher during high water level with sufficient substrate and anaerobic environment (Fig. 6b). In particular, the contents of mud flat and nymphoides

zone increased most during flooding. The anammox was also promoted by the water-solid interface under anaerobic conditions in the absence of oxygenated macrophytes. In wetlands, organic matters promoted the large number of denitrifying bacteria and anammox bacteria to compete for $NO_2^-$ (Guven et al., 2005; Tang et al., 2010). With the input of organic matters to sediments caused by flooding, the increase of denitrifying functional genes in each vegetation zone was higher than that of anammox functional gene (Fig.6b), which reflected the competition for substrate in the two microbial processes. However, due to the

complex process of nitrogen cycling in wetlands, further studies were needed to expound the competitive relationship between denitrification and anammox.

## 5 Conclusions

As Dongting Lake maintains a natural connection with the Yangtze River and it is located at the lower reaches of the Three Gorges Dam, the hydrological conditions of Dongting Lake are affected by human regulation. Hydrological conditions (time

and intensity of flooding) have significant effects on the stratification and distribution of vegetation, and the hydrological conditions are important in establishing the vegetation communities. Long-term flooding causes the vegetation to wither and fall, which makes the sediment a source of organic matter. In contrast, the sediment provides many elements for the growth of vegetation during low water levels. Thus, the contents of various ecological factors in sediments differ during different water





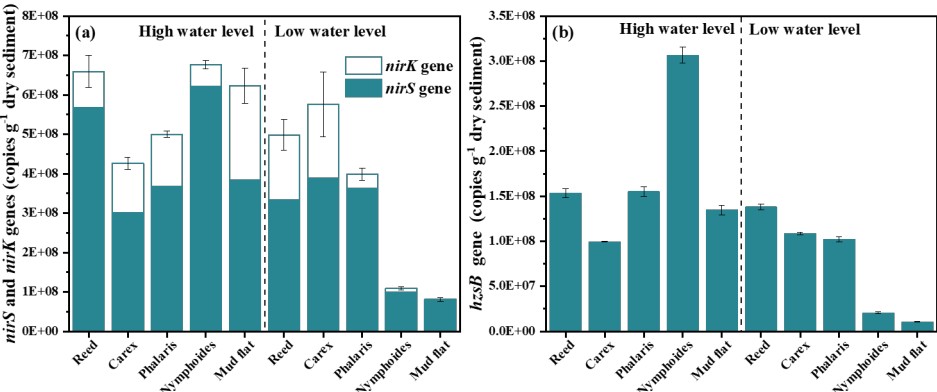

**Figure 6.** Abundance of denitrifying *nirS* and *nirK* genes (a) and anammox hzsB gene (b) in sediments during different water levels

levels. In this process, the characteristics of different vegetation led to differences in the ecological factors in the sediments

of different vegetation zones. The C:N ratio of sediments is an important factor of $N_2O$ emission and microbial activities in denitrification. Significant correlations between $N_2O$ emission flux and the C:N ratio of sediments were observed. The $N_2O$ emission flux decreased as the C:N ratio of sediments decreased. Moreover, differences were observed in the denitrifying functional genes (*nirS* and *nirK*) and anammox functional gene (*hzsB*) in the sediments of different vegetation zones and during different water level periods. This further verifies the differences in the $N_2O$ emission flux. Nevertheless, denitrification

is one of the pathways to produce $N_2O$, while the changes in ecological factors of sediments and hydrological conditions also affect other microbial processes that produce $N_2O$. Future research should focus on the effect of changes in the vegetation under hydrological stress on sediment and the complete nitrogen-cycle of the sediment in the lakeshore zone. The results of these studies may significantly help to optimize the water conservancy regulations and research of wetland biodiversity.

*Author contributions.* Huazu Liu: Investigation, Formal analysis, Writing - Original Draft; Qiu Jin: Investigation, Formal analysis, Writing

- Original Draft; Ruijie Shi: Investigation, Formal analysis, Methodology; Chengxu Lv: Investigation, Methodology; Junxiao Luo: Investigation, Methodology; Yan He: Investigation, Formal analysis; Wei Yang: Investigation, Formal analysis; Xiaoguang Xu: Conceptualization, Investigation, Supervision; Shenghua Qian: Writing - Review & Editing, Supervision; Wei Li: Conceptualization, Writing - Review & Editing, Supervision.

*Competing interests.* The authors declare that they have no known competing financial interests or personal relationships that could have

appeared to influence the work reported in this paper.





*Acknowledgements.* The authors acknowledge the administration of West Dongting Lake National Nature Reserve, China for assistance in the sampling. This work was supported by the National Natural Science Foundation of China (31700401 and 51809180), the Fundamental Research Funds for the Central Universities (2019CDQYCH014 and 2020CDJQY-A014), and the Scientific Research and Technology Development Program of Guangxi (2018AB36010).



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
