# Peer review of "Effect of vegetation distribution driven by hydrological fluctuation on sedimental stoichiometry regulating $N_2O$ emissions in freshwater wetland"

_Biogeosciences, 2021_

## Author Comment (AC1)

**Reply on RC1**

Huazu Liu et al.

**RC1**

Overall, the idea of the study is interesting, however the major limitation is that it is based on only few grab samples (both soil and gas). Samples were collected once during low water level event and once during high water level event. Making concluding about the ecosystem based on few samples is not sufficient. For example, Figure 1 shows regression analyses that is based on only few points and same goes with other figures as well. To understand the dynamics at an ecosystem level, a much larger amount of samples should be collected. First to see the seasonal dynamics and secondly to have a realizable amount of data for statistical analyses.

**Author's response:** We thank the anonymous referee#1 for the detailed reviews with relevant and constructive comments to improve the quality of the manuscript. The received recommendations were carefully considered and incorporated into the current version of the manuscript. We focused on the differences between nitrogen cycle during high and low water levels in ecosystem. Therefore, we monitored $N_2O$ emissions, vegetations and soil in the steady period after the change of water level. In addition, influenced by the Three Gorges Dam, the annual change of water level in the study area were very regular. As a result, the variations between years in the change of water level were very small. Although we set up three sampling sites at each vegetation zone in the study area, we collected as many samples as possible at each sampling site for statistical analysis. And we ensured that the distances between the sampling sites made less interference between the sites. We agreed the detailed review and comments which will be helpful in our future researches. A point-by-point response to comments was given below.

- Figure 1 - Photos have low quality. Location of the region would be nice to show.

**Author's response:**

Thanks for the correction; we have replaced a photo with high quality.

- Lines 115-120 - You inserted pedestal into the soil and then started to collect gas samples. How long was the stabilisation period because this could create relatively large distrubance to the soil? How many gas samples were used to calculated flux? How did you access the site during high flood to avoid soil distrubance? The size of the chambers?

**Author's response:**

For the first sampling, we inserted the pedestal about 10cm into the soil. And we didn't take the pedestal to reduce the disturbance of subsequent samples. After the pedestal into the soil, we set a stabilization period of 30 minutes and a board to reduce soil disturbance from people (Wang, H. J., Wang, W. D., Yin, C. Q., Wang, Y. C., and Lu, J. W.: Littoral zones as the "hotspots" of nitrous oxide ($N_2O$) emission in a hyper-eutrophic lake in China, Atmospheric Environment, 40, 5522-5527, 10.1016/j.atmosenv.2006.05.032, 2006.)(as shown in the figure below).

[Figure]

We shut down the ship's machinery in the study area and waited for an hour before sampling during high water level. Then, we let the chamber on the water for 30 minutes before collecting the gas. We used a 10m air pipe, which kept chamber as far away from the ship as possible to reduce disturbance (as shown in the figure below).

[Figure]

Seven gas samples were used to calculated flux in each sampling site. And three sampling sites were set in each vegetation zones.

[Figure]

We described the size of the chambers in the manuscript as following:

L116-117. "The volume of the upper chambers used during low water level was 0.028 m$^3$ ($h$ = 40cm, $\Phi$ = 30cm), and the volume of the pedestal was 0.011 m$^3$ ($h$ = 15cm, $\Phi$ = 30cm). And the volume of the chambers used during high water level was 0.018 m$^3$ ($l$ = 40cm, $w$ = 30cm, $h$ = 15cm)."

- Line 135 - Statistical analyses: was the data normally distributed? And what tests were used to control that?

**Author's response:**

We used KS-test to confirm that the data was normally distributed (as shown in the figure below).

[Figure]

- Figure 3 - caption is not referring to correct sub-plots. E.g. B is TOC not nitrogen density etc.

**Author's response:**

New caption was as following:

L171-172. "Figure 3. Content of carbon and nitrogen in vegetation and sediments during different water levels. Carbon (a) and nitrogen (c) densities of vegetation in different zones. Concentration of TOC (b) and TN (d) in sediments in different vegetation zones"

- Figure 5 - text in the figure is so small that it is unreadable.

**Author's response:**

We have resized the text in the figure as suggest.

- Line 350 - do you have data about N$_2$O reducers: *nosZ* clade I and II genes? Currently the abundance of *nirS*, *nirK* and *hzsB* genes does not provide enough information about the entire N cycle.

**Author's response:**

In this study, we mainly focused on N$_2$O emissions in the N cycle. Nitrite is converted to NO or N$_2$O by nitrite reductase (NIR) in denitrification, the extensively used biomarkers for which are nirK (Cu-containing) and nirS (cytochrome cd 1) (Levy-Booth, D.J., Prescott, C.E., Grayston, S.J.: Microbial functional genes involved in nitrogen fixation, nitrification and denitrification in forest ecosystems, Soil Biol. Biochem., 75, 11–25, 10.1016/j.soilbio.2014.03.021, 2014.). And the N$_2$O emission varied with the abundance of *nirS* and *nirK* genes (Zhang, L., Jiang, M.H., Ding, K.R., Zhou, S.G., Iron oxides affect denitrifying bacterial communities with the nirS and nirK genes and potential N$_2$O emission rates from paddy soil, EUROPEAN JOURNAL OF SOIL BIOLOGY, 93, 103903, 10.1016/j.ejsobi.2019.103093, 2019). Thus, the abundance of *nirS* and *nirK* genes became the main object of discussion. Meanwhile, in order to further explore the N cycle in the anaerobic environment such as reservoirs and lakes, we analyzed the functional gene (*hasB* gene) of anammox to compare with the denitrification.

- Throughout the text: sometimes N2O has subscript (N$_2$O) and sometimes not.

**Author's response:**

We double checked the subscripts and revised the incorrect subscripts.

---

## Author Comment (AC2)

**Reply on RC2**

Huazu Liu et al.

**RC2**

**A:**

The manuscript submitted by Liu and colleagues investigates relationships between plant species, hydrology and N2O fluxes. In their work, they evaluate four (or five?) vegetation zones in a Chinese wetland and analysed C and N contents in the vegetation and sediments, N2O fluxes, microbial communities and selected genes involved in the N cycle during high and low water levels. They conclude that the distribution of plants under hydrological conditions modified the stoichiometric ratio of sediments, resulting in the variations of N2O emission fluxes and microbial communities in the vegetation zones.

While the topic is interesting and relevant for the journal, I have my serious doubts about the experimental design and the approach used. One of the main arguments of the manuscript is that the vegetation distribution is driven by hydrological changes; it is also argued that is the vegetation distribution the factor affecting the emission of N2O (Abstract, L3). Your first objective was indeed to examine the relationship between hydrology and species distribution. I was however not able to understand how your experimental set up was helpful to elucidate more about this matter, and which kind of data you use to support that this is indeed the case in your plots. You merely monitored the water level across the vegetation types and, actually, found that all vegetation types except reed were having exactly the same pattern (Figure 2a). And, even if you find a distinct pattern in the water dynamics across your vegetation zones, you won't be able to conclude whether if it is the hydrology or the plant communities the ones driving the N2O fluxes.

**Author's response:**

We would like to thank anonymous referee#2 for the detailed reviews with relevant and constructive comments to improve the quality of the manuscript. The received recommendations were carefully considered and incorporated into the current version of the manuscript.

Hydrological conditions, such as flooding time, flooding depth, and flooding frequency, were the dominant factors driving vegetation distribution (Tan, Z. Q., Zhang, Q., Li, M. F., Li, Y. L., Xu, X. L., and Jiang, J. H.: A study of the relationship between wetland vegetation communities and water regimes using a combined remote sensing and hydraulic modeling approach, Hydrol. Res., 47, 278-292, 2016.; Toogood, S. E., Joyce, C. B., and Waite, S.: Response of floodplain grassland plant communities to altered water regimes, Plant Ecology, 197, 285-298, 2008.). Plant species had different amounts of carbon and nitrogen in their organisms (Elser, J. J., Fagan, W. F., Denno, R. F., Dobberfuhl, D. R., Folarin, A., Huberty, A., Interlandi, S., Kilham, S. S., McCauley, E., Schulz, K. L., Siemann, E. H., and Sterner, R. W.: Nutritional constraints in terrestrial and freshwater food webs, Nature, 408, 578-580, 2000.; Yu, Q., Chen, Q., Elser, J. J., He, N., Wu, H., Zhang, G., Wu, J., Bai, Y., and Han, X.: Linking stoichiometric homoeostasis with ecosystem structure, functioning and stability, Ecology Letters, 13, 1390-1399, 2010.). Therefore, we hypothesized that the biomass (carbon density and nitrogen density) of plants can be used as an index to distinguish plant species. And the change rate of water level (Figures 2a and 2b) can reflect the flooding time, frequency and depth. Thus, we explored the driving effect of hydrology on vegetation distribution by analyzing the relationship between the biomass (carbon density and nitrogen density) of plants and the change rate of water level. In fact, there were differences in the water level of all vegetation zones. Due to the large flooding depth, the difference in water level changes of these vegetation zones was not obvious in Figure 2a. In order to verify this, we scaled the image and showed part of the data during high water level (as shown in the figure below).

We thought that hydrology drove vegetation distribution, and the vegetation distribution effected the content of carbon and nitrogen in soil. The changes in the content of carbon and nitrogen (or stoichiometric proportion) varied the microbial processes and N2O emission. As anonymous referee#2 said, there were many factors affecting N2O emission, and we won't be able to conclude whether if it is the hydrology or the plant communities the ones driving the N2O fluxes. However, our study attempted to reveal that the change of water level mediated by plant community drove the N2O emission in wetlands. Additionally, we also discussed the influence of hydrology, plant communities and other factors on N2O emission in the section of *Discussion*.

Further, you focus on N2O emissions. We know that the temporal and spatial variability of N2O fluxes can be really high so I strongly suspect that measuring only twice a year over three (pseudo?) replicates is not enough to adequately catch the dynamics of the fluxes, especially with this highly contrasting environmental conditions. I also see that you measured on the soil surface (during low water level conditions) and on the water surface (high water level). This means your measurement conditions are totally different (different chamber setup, different diffusion coefficients, etc). I miss a clear explanation on how the different measurement conditions may have affected your results. For example, if I interpret Fig S2 correctly, I can see that the starting concentrations when setting the chamber (which should be the atmospheric N2O concentration) differ by a factor of three, which is hard for me to understand.

**Author's response:**

We focused on the differences between nitrogen cycle during high and low water levels in ecosystem. Therefore, we monitored N2O emissions, vegetations and soil in the steady period after the change of water level. In addition, influenced by the Three Gorges Dam, the annual change of water level in the study area were very regular. As a result, the variations between years in the change of water level were very small. Although we set up three sampling sites at each vegetation zone in the study area, we ensured that the distances between the sampling sites made less interference between the sites. The approximate location of sampling sites was shown in the figure below, and the yellow box represents sampling sites.